# Protein polarization driven by nucleoid exclusion of DnaK(HSP70)–substrate complexes

Clémence Collet[1,2,3,4], Jenny-Lee Thomassin[1,2,3,4], Olivera Francetic[5], Pierre Genevaux[6] & Guy Tran Van Nhieu[1,2,3,4]

Many bacterial proteins require specific subcellular localization for function. How *Escherichia coli* proteins localize at one pole, however, is still not understood. Here, we show that the DnaK (HSP70) chaperone controls unipolar localization of the *Shigella* IpaC type III secretion substrate. While preventing the formation of lethal IpaC aggregates, DnaK promoted the incorporation of IpaC into large and dynamic complexes (LDCs) restricted at the bacterial pole through nucleoid occlusion. Unlike stable polymers and aggregates, LDCs show dynamic behavior indicating that nucleoid occlusion also applies to complexes formed through transient interactions. Fluorescence recovery after photobleaching analysis shows DnaK-IpaC exchanges between opposite poles and DnaKJE-mediated incorporation of immature substrates in LDCs. These findings reveal a key role for LDCs as reservoirs of functional DnaK-substrates that can be rapidly mobilized for secretion triggered upon bacterial contact with host cells.

[1] Equipe Communication Intercellulaire et Infections Microbiennes. Centre de Recherche Interdisciplinaire en Biologie (CIRB). Collège de France, 11, Place Marcelin Berthelot, 75005 Paris, France. [2] Institut National de la Santé et de la Recherche Médicale (Inserm) U1050, Paris, Cedex 15, France. [3] Centre National de la Recherche Scientifique (CNRS) UMR7241, 75016 Paris, France. [4] MEMOLIFE Laboratory of excellence and Paris Science Lettre, Paris, Cedex 15, France. [5] Biochemistry of Macromolecular Interactions Unit, Institut Pasteur, Department of Structural Biology and Chemistry, CNRS UMR3528, 28 rue du Dr Roux, 75724 Paris, Cedex 15, France. [6] Laboratoire de Microbiologie et de Génétique Moléculaires, Centre de Biologie Intégrative (CBI), Université de Toulouse, CNRS, 118 route de Narbonne, 31062 Toulouse, Cedex 9, France. These authors contributed equally: Clémence Collet, Jenny-Lee Thomassin. Correspondence and requests for materials should be addressed to G.T.V.N. (email: guy.tran-van-nhieu@college-de-france.fr)

In bacteria, protein sub-localization plays a key role in their function and regulation[1–3]. Unlike eukaryotic cells, there are no intracellular membranes delimiting internal compartments in bacteria suggesting localization relies on other mechanisms[2]. Mechanisms describing "functional" cytoplasmic protein localization at bacterial poles remain controversial, since amorphous aggregates are excluded from the nucleoid and accumulate at the poles or forming septa[4–6]. "Nucleoid occlusion", coined to describe how the nucleoid prevents FtsZ division ring formation in DNA-rich areas[7], describes an entropy-driven process whereby nucleoid molecular crowding and diffusion hindrance direct protein aggregates into nucleoid-free regions[8]. Small protein aggregates appear to act as seeds feeding the amorphous growth of larger aggregates, explaining unipolar accumulation[6], yet high rates of protein addition favor bipolar localization[9]. More controversy arose from studies showing protein fusions with fluorescent reporters can result in polar localization[10]. Yet "functional" proteins also accumulate at bacterial poles, a feature critical to their function[2,11,12]. Two main mechanisms have been proposed for polar protein complex formation. In the "diffusion-capture" mechanism, proteins are targeted through interactions with a polar localization determinant. This mechanism has been proposed for proteins involved in cell division and chromosome segregation in *E. coli*, cell cycle control in *Bacillus subtilis* and *Caulobacter crescentus* (reviewed in ref.[2]). In many cases, the primary polar localization determinant remains elusive[13]. In *B. subtilis* the self-oligomerizing protein DivIVA localizes to the pole by sensing increasing negative membrane lipid curvature[14]. Alternatively, for a few proteins, polar localization can result from protein binding to cardiolipin, a phospholipid that forms lipid microdomains at the pole[15]. The second proposed mechanism for polar localization is nucleoid occlusion. A well-studied example is the cell cycle regulator protein PopZ in *C. crescentus* that forms polar multimeric scaffolds[16]. PopZ self-oligomerization leads to the formation of matrix of selective permeability in regions with low DNA density, such as forming septa or poles. PopZ polar localization is maintained in *E. coli* and depends on its self-oligomerization domain[16].

Various surface proteins have unipolar localization[17]. A well-known example in *E. coli*-related bacteria is the *Shigella* IcsA autotransporter protein responsible for intracellular actin-based motility. In contrast to *Listera* ActA that is secreted before moving to the pole through diffusion and accumulation of old cell wall material, cytoplasmic IcsA is targeted to the pole prior to secretion[18] through mechanisms that are not fully understood. Genetic approaches failed to identify IcsA polarization determinants and showed polarity was independent of FtsZ, the Min system and nucleoid occlusion[12]. However, these approaches cannot identify essential genes. A genome wide screen of proteins fused to green fluorescent protein (GFP) implicated the DnaK chaperone in IcsA polarization, but how this chaperone controls IcsA polar localization remains unclear[12].

The *Shigella* type III translocon component IpaC localizes to one bacterial pole prior to cell contact and determines polar type III secretion during host cell invasion by *Shigella*[11]. IpaC unipolar localization parallels that of IcsA, and does not result from protein aggregation since tagged proteins are chased from the pole by endogenously produced proteins[11]. Here, we identified proteins involved in IpaC polarization. We found that direct and reversible IpaC-DnaK interactions as part of large dynamic complexes (LDCs) drive polarization through nucleoid occlusion. Polar accumulation was associated with DnaKJ-dependent substrate folding.

## Results

### IpaC and DnaK form insoluble complexes upon cross-linking.
Previous work identified IpaC regions involved in polarization; the U region is dispensable, whereas others including the C-terminal self-oligomerization region are essential[11]. To identify IpaC polarization determinants, we performed pull-down experiments with a hexahistidine-tagged construct of IpaC lacking the U region fused to GFP (CiΔU-His) that localizes to the pole and is produced at high levels (Fig. 1a, b). No unique CiΔU-His interaction partners were identified relative to a GFP control,

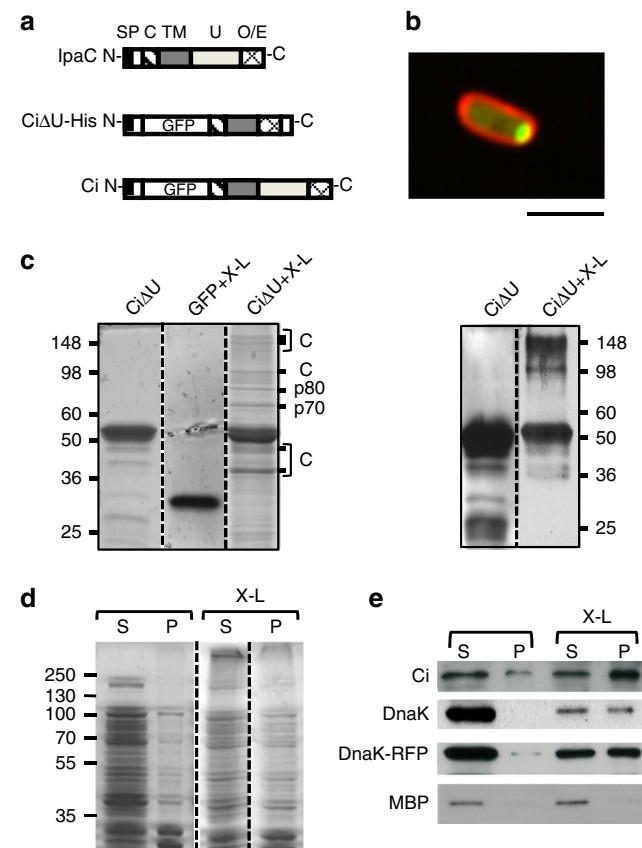

**Fig. 1** IpaC binds to DnaK and forms insoluble complexes upon cross-linking. **a** IpaC and CiΔU-His domain organization. SP: type III secretion signal peptide between residues 1–20; C: IpgC chaperone binding site between residues 50–80; TM: putative transmembrane domain between residues 100–170; U: domain with unknown function between residues 170–301; O/Ef: oligomerization/effector domain between residues 302–363. CiΔU-His is a polar derivative of IpaC, containing a deletion of residues 101–302, an insertion of GFP at residue 57, and a hexahistidine tag at its C-terminus. **b** Representative fluorescence micrograph of MC4100/pCi. Red; FM4-64 membrane staining, Green: Ci fluorescence. Scale bar = 2 μm. **c** Left panel: Coomassie blue staining following SDS-PAGE analysis of proteins associated with GFP and CiΔU in pull-down experiments from bacterial lysates. C indicates Ci related proteins and p70 and p80 indicate protein species at 70 and 80 kDa, respectively. Right panel: anti-IpaC western blot analysis. The molecular weight standards are indicated in kDa (left). **d** Coomassie blue staining of soluble (S) and insoluble (P) fractions of lysates from *E. coli* MC4100/pSUCi/pDnaK-RFP. X-L: cross-linked samples. **e** After treatment for 10 min at 100 °C to dissociate complexes and SDS-PAGE, western-blot analysis was performed using antibodies against IpaC (Ci), DnaK, or the periplasmic maltose binding protein (MBP) as a control, of filter replicas corresponding to D. **c**–**e** Each lane corresponds to loads normalized to equivalent bacterial $OD_{600 \text{ nm}}$, with S and P resuspended in identical volumes

and only smaller IpaC-cross reacting peptides were observed (Fig. 1c). Therefore, protein complexes were stabilized by formaldehyde cross-linking prior to lysate preparation and pull-down experiments. After cross-linking, most of the CiΔU-His was insoluble (Fig. 1d). However, some CiΔU-His and several CiΔU-His associated proteins were visualized after affinity chromatography and SDS-PAGE analysis (Fig. 1d). Western blot and mass spectrometry analyses detected IpaC oligomers and degradation products (Fig. 1e). In addition, two major protein species co-eluted with CiΔU-His; mass spectrometry analysis identified the 70 and 80 kDa proteins as DnaK and elongation factor G, respectively (Fig. 1c). The link between DnaK and IcsA localization[12] prompted us to characterize the IpaC-DnaK interaction.

**Large IpaC-DnaK complexes are observed at the bacterial pole**. To characterize DnaK-IpaC complexes, bacterial lysates were fractionated. As expected from pull-down experiments, full-length IpaC fused to monomeric GFP (Ci) and DnaK were mainly recovered in the soluble fraction. In addition, no interaction between Ci and DnaK was detected, unless cross-linking was performed (Fig. 1d, e, and Supplementary Fig. 1a). Upon cross-linking, large amounts of DnaK and Ci were in the insoluble fraction (Fig. 1e). DnaK-RFP that was mostly soluble in the absence of cross-linker also became insoluble upon cross-linking (Fig. 1e). Insoluble fractions had high protein complexity (Supplementary Figs. 1c and d), consistent with the reported DnaK interactome and with its co-translational binding to nascent polypeptides[19].

The insolubility of cross-linked Ci and DnaK at low centrifugation forces suggests they form large complexes hereafter called LDCs. These complexes are similar in size to inclusion bodies resulting from misfolded protein aggregation, with the important difference that aggregates form irreversibly without cross-linking. Given the similarity, a method to purify recombinant aggregated proteins from inclusion bodies was used to purify LDCs isolated after cross-linking[20]. Lysates were prepared from an *E. coli* strain co-producing fluorescently tagged Ci and DnaK-RFP, and purified LDCs were analyzed by fluorescence microscopy. Ci and DnaK-RFP containing LDCs form particles with diameters ranging from ca. 700 nm to the $X–Y$ resolution limit (200–300 nm), similar to polar structures observed in bacteria (Fig. 1e and Supplementary Fig. 1b). SDS-PAGE analysis of total proteins indicated that DnaK-RFP and Ci were enriched in LDCs (Supplementary Figs. 1c and d). These results indicate that transient Ci-DnaK interactions result in LDC formation at the bacterial pole.

**LDCs form a DnaK-dependent selective permeability matrix**. *C. crescentus* PopZ self-oligomerizes to form a selective permeability matrix that is confined to the bacterial pole through nucleoid occlusion, a phenotype that is recapitulated in *E. coli*[16,21,22]. The PopZ oligomerization domain is required, but not sufficient for polar localization[2]. Similarly, the IpaC oligomerization domain is required but not sufficient for polar localization[11]. We hypothesized that IpaC-DnaK interactions result in matrix formation, since DnaK forms oligomers, particularly under its ADP-bound conformation[23,24].

Consistent with previous reports in bacteria grown at 30 °C, DnaK-RFP had diffuse localization and bipolar localization during growth at 37 °C (Fig. 2a, b)[12,25]. In contrast, Ci was primarily unipolar regardless of growth temperature (Fig. 2b, c). Strikingly, Ci production shifted DnaK-RFP localization to one pole, even at 30 °C (Fig. 2b). The Ci-related change in DnaK localization at 30 °C correlated with increased DnaK and GroEL

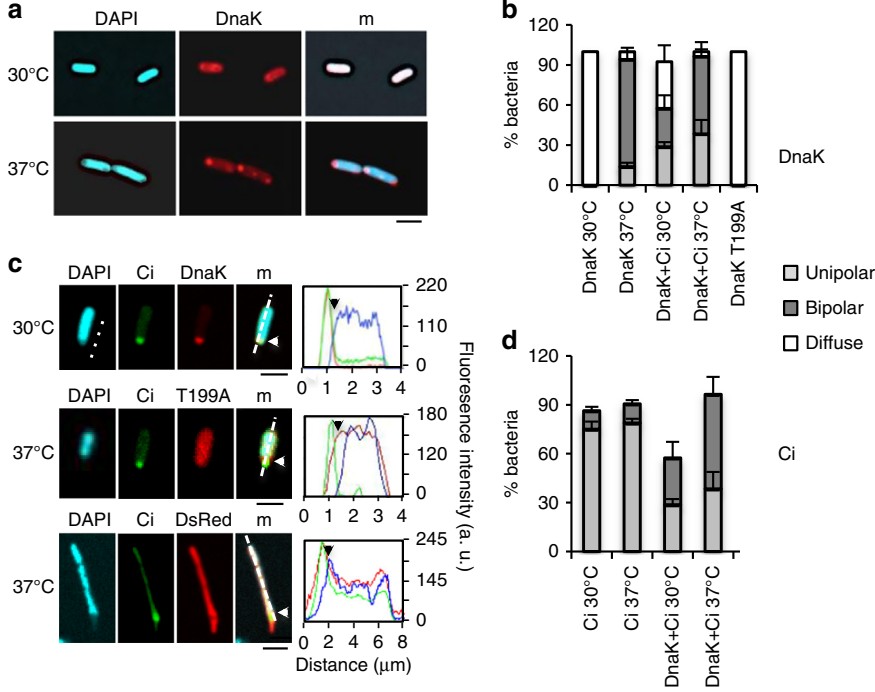

**Fig. 2** Ci and DnaK-RFP show an interdependent recruitment at LDCs. Bacteria were grown to exponential phase in the presence of 22 mM glucose at the indicated temperature. Induction of DnaK-RFP and Ci was performed by addition of IPTG and arabinose at a final concentration of 100 µM and 2.2 mM, respectively, and incubation for 60 min. Samples were fixed and analyzed by fluorescence microscopy. **a**, **c** Representative confocal micrographs of MC4100/pDnaK-RFP (**a**), or MC4100/pSU-Ci co-expressing pDnaK-RFP (DnaK), pDnaK-T199A-RFP (T199A) or DsRed4T (DsRed) (**c**). Line scan analysis corresponding to the dotted lines in DnaK-RFP or DnaK-RFP T199A with the corresponding colors (**c**, left). DnaK-RFP; red: Ci; green: DAPI: cyan. Scale bar = 2 µm. White and black arrows indicate the edge of discrete Ci foci in bacteria. Percent of bacteria (Mean ± SEM) with diffuse, bipolar or unipolar localization of DnaK-RFP (**b**) or Ci (**d**) ( > 200 bacteria, N = 2)In figure 2 panel "d" is present in the figure legend but it is not present un the figure please provide.I uploaded a new version of figure 2 with panel d clearly indicated

levels, resembling a heat-shock response (Supplementary Fig. 2). Polar DnaK-RFP accounted for 55.1 ± SEM 0.6% of the total pool of DnaK-RFP, compared to 18.7 ± 0.5% in the absence of Ci. While Ci was mostly unipolar, its localization became bipolar in the presence of DnaK-RFP (Fig. 2b), suggesting increased DnaK-levels affect LDC formation. ATP-bound DnaK initially encounters substrates and ATP hydrolysis results in the formation of stabilized ADP–DnaK–substrate complexes[26]. Consistent with Ci association with ADP–DnaK at the bacterial pole, a DnaK (T199A) variant locked in an ATP-bound conformation did not co-localize with Ci (Fig. 2b, c)[27]. Fluorescence intensity measurements along the bacterial axis confirmed accumulation and co-localization of Ci and DnaK-RFP at the bacterial poles, consistent with complex formation, and mutual exclusion of nucleoid and LDCs (Fig. 2c). Unlike the nucleoid, controls DsRed and DnaK-RFP T199A were not excluded from LDC-containing poles (Fig. 2c) indicating that LDCs form a matrix of selective permeability. These results suggest LDCs are soluble and reversible complexes and not protein aggregates.

**Lethal Ci aggregates are formed in the absence of DnaKJ.** DnaK association with its co-chaperone DnaJ is essential for the recognition of stress-induced substrates, or substrates implicated in the DNA replication initiation complex. DnaJ stimulates ATP hydrolysis on DnaK, enabling conformational changes in the ADP-bound DnaK substrate-binding domain that stabilize substrate association and promote DnaK oligomerization[23,28,29]. In ΔdnaK, some bacteria retain Ci unipolar localization (21 ± SEM 2.2%), whereas in the majority of bacteria Ci localization was altered, with most (71 ± 2.2%) bacteria containing several Ci patches and some bacteria (6.6 ± 0.1%) were Ci-filled (Fig. 3b). Similar Ci localization patterns were observed in strains lacking DnaJ or the DnaJ, CbpA and DjlA co-chaperones (ΔJ[3]), with most containing multiple Ci patches (Fig. 3a, b). This was particularly striking in ΔdnaKJ, where 33 ± 7.4 % of bacteria were filled with Ci (Fig. 3a, b). Fractionation experiments indicated that in the absence of DnaK or its co-chaperones, patches were associated with decreased Ci solubility, consistent with a role for the DnaK machinery in preventing Ci aggregation (Fig. 3c, d).

DnaK is essential for bacterial growth at temperatures above 37°C. Ci expression in ΔdnaKJ, but not in the wild-type strain, severely inhibited bacterial growth, even at the permissive temperature of 30 °C (Supplementary Fig. 3a). Accordingly, Ci expression in ΔdnaKJ induced the formation of filamentous bacteria with large Ci patches. These patches were not restricted to the bacterial poles, expanded during the induction period (Fig. 3a and Supplementary Fig. 3a), and remained mutually exclusive with the nucleoids. Bacteria containing large patches of Ci with no detectable DNA, or sometimes bacteria devoid of nucleoid were observed (Fig. 3a).

Together, these results indicate that DnaKJ is essential for the folding or maintenance of a soluble pool of Ci and for its polarization as part of large reversible complexes. In the absence of DnaKJ, Ci aggregates appeared to prevent nucleoid expansion and inhibit cell division.

**The nucleoid restrains Ci and DnaK at the bacterial pole.** Nucleoid occlusion relies on the mutual exclusion of polymers; the bacterial chromosome excludes polymers formed by proteins assembled into large matrices or aggregates[30]. While there is mounting interest in the role of transient interactions in defining bacterial spatial networks, little is known about polar confinement of reversible complexes by the nucleoid. Such considerations are not intuitive, as this entropy-driven segregation mechanism in E.

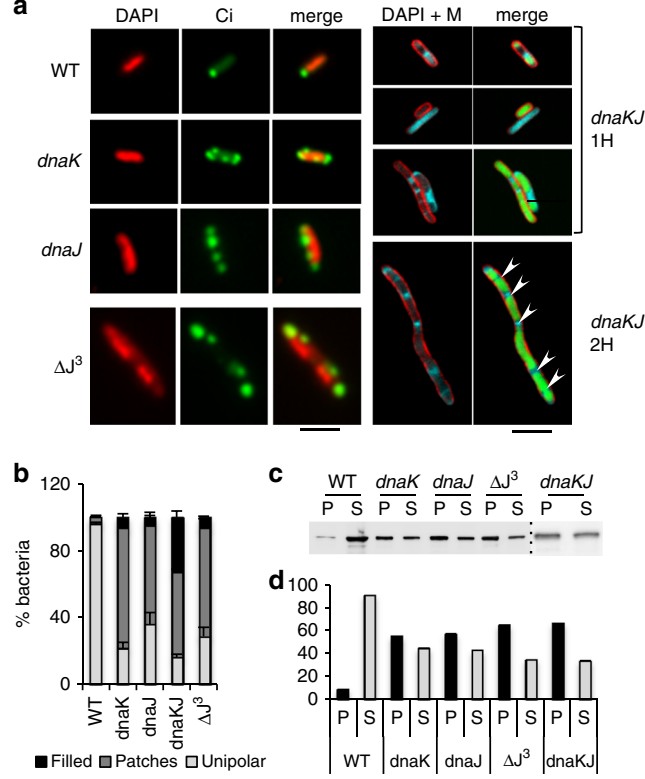

**Fig. 3** DnaKJ prevents the formation of Ci lethal aggregates. **a** Ci expression was induced in the indicated strains. ΔJ[3]: dnaJ cbpA djlA mutant. Panels on the right show dnaKJ mutants grown for 1 h (1H) or 2 h (2H). Representative micrographs of fixed samples processed for fluorescent staining of DNA (DAPI), and bacterial membranes (M). Green: Ci-GFP fluorescence. Scale bar = 2 μm. **b** Percent of bacteria (Mean ± SEM) with filled (black), unipolar (gray), or patches (dark gray) of Ci fluorescence ( > 35 bacteria, N = 4). **c** Anti-IpaC western blot analysis of soluble (S) and insoluble fractions (P) prepared from lysates of the indicated strains. **d** Quantification of Ci band intensities in **c** by scanning densitometry

coli is most effective for large complexes with diameters above 50 nm[31], much larger than expected for DnaK–Ci complexes.

The nucleoid and LDCs in bacteria were visualized by fluorescence microscopy and the average signal of foci was plotted relative to the bacterial mid-cell, as described previously[32,33]. LDCs were detected at the bacterial pole away from the mid-cell and outside the nucleoid-containing region (Fig. 4a, b). These data clearly indicate that LDCs and the nucleoid are spatially excluded.

LDC assembly and disassembly was analyzed by time-lapse microscopy (Fig. 4c–i). Tracking experiments showed that LDCs appeared and grew at the bacterial poles (Fig. 4c–e, Supplementary Movie 1). This is consistent with LDC formation at the pole and DnaK binding neosynthesized Ci prior to incorporation into the growing LDC. In the absence of Ci induction, LDCs remained at the bacterial pole and the signal decreased below the detection limit (Fig. 4f–h, Supplementary Movie 2), consistent with previously observed Ci turnover in LDCs[11]. In contrast, in the absence of induction supplementation with 2,4-dinitrophenol (DNP), an ATP uncoupler that "glassifies" the cytoplasm by reducing metabolic activity and increasing viscosity[33], stabilized LDCs at the pole with no decrease in signal, even after prolonged incubation (Fig. 4f–h, Supplementary Movie 3). The number and location of LDCs in DNP-treated cells remained unchanged, even when incubation times were increased to 1 h (Fig. 4i).

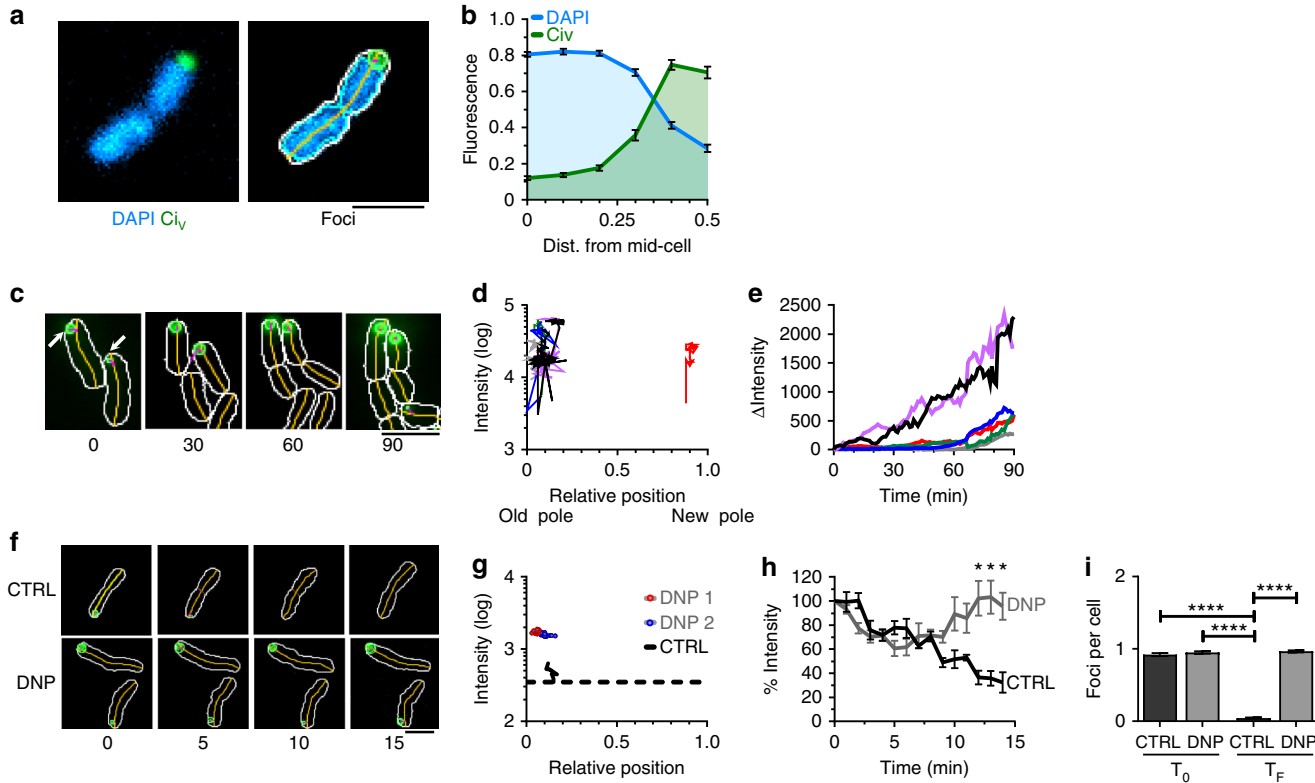

**Fig. 4** LDCs appear and remain at the bacterial pole. **a**, **b** Distribution of LDCs along the cell major axis. Green: $Ci_v$; cyan: DAPI. **a** (left) Representative micrographs of bacteria with DAPI and Ci fluorescence, shown are cell outline, mid-line and foci detected using MicrobeJ software. **b** Mean ± SEM of the fluorescence intensity relative to the cell major axis (106 bacteria, $N = 3$). **c–i** Live cell fluorescence microscopy analysis of assembly and disassembly of LDCs. **c**, **f** Stills from time-lapse series, with the elapsed time indicated in min. **d**, **g** Tracks of LDCs' peak intensity of foci along the bacterial major axis (shown are representative tracks). Dashed line indicates limit of detection. **e** Traces: fluorescence intensity of single LDCs corrected for initial intensity plotted over time (Shown are traces from six individual bacteria). **h** Mean ± SEM of LDC fluorescence intensity normalized to initial fluorescence (% Intensity; 1 field of view (six bacteria), $N = 3$). Holm-Sidak test, statistical significance relative to untreated bacteria, asterisk: $p < 0.05$. **i** Mean ± SEM of the number of foci in the presence or absence (CTRL) of DNP at the onset of analysis ($T_0$) or following 60 min incubation ($T_F$) (foci were counted in 278, 352, 1096 and 486 bacteria from CTRL and DNP ($T_0$), CTRL and DNP ($T_F$), respectively; $N = 3$). Statistical significance determined with Kruskal–Wallis test followed by Dunn's post hoc analysis, four asterisks: $p < 10^{-4}$. **d**, **e** MC4100/pCi$_v$. **f–i** MC4100/pSUCi. Arrows: LDCs. Scale bar = 2 µm

To clarify the role of nucleoid occlusion in LDC confinement to the bacterial pole, DNA in LDC-containing bacteria was degraded using the bacteriocin Colicin E2[34]. Colicin E2 treatment resulted in DNA degradation within 15 min, with extensive DNA-degradation after 2 h (Supplementary Fig. 4a). When Colicin E2 treatment was analyzed by DAPI staining and fluorescence microscopy, decreased DNA signal in the area extending from the bacterial poles and dense residual DNA signal was observed at mid-cell (Supplementary Fig. 4, Supplementary Fig. 5a and Supplementary Movie 5). Consistent with the nucleoid preventing LDC diffusion, in Colicin E2 treated cells Ci relocalized to the extended area devoid of DNA at both cell poles, but was still excluded from the remaining nucleoid at mid-cell (Supplementary Fig. 5b). Colicin E2 treatment led to a 2.9 fold increase in the Ci area size compared to untreated bacteria, with an average size of $1.2 \pm$ SEM 0.8 and $0.5 \pm 0.2$ µm$^2$ for Colicin-treated and control cells, respectively (Supplementary Fig. 5d). Conversely, the average fluorescence intensity decreased by about two-fold in Colicin-treated cells relative to untreated cells (Supplementary Fig. 5c), consistent with Ci redistribution into DNA-free areas. In contrast, Ci patches formed in $\Delta dnaKJ$ did not redistribute into DNA-free regions induced by Colicin E2, consistent with the absence of diffusion observed for aggregates (Supplementary Fig. 5e)[5,6].

Similar results were obtained after ciprofloxacin treatment, a quinolone that inhibits DNA gyrase bound to DNA, causing DNA double strand breaks and degradation[35]. As for Colicin E2, ciprofloxacin treatment induced the formation of extended polar areas devoid of DNA, filled with Ci (Supplementary Figs. 5b, 6). Of note, ciprofloxacin treated cells also showed fusiform extended poles filled with Ci (Supplementary Fig. 7). Ciprofloxacin-treatment led to a 3.4-fold increase in Ci area compared to untreated bacteria, with an average area of $1.8 \pm 0.8$ µm$^2$ for ciprofloxacin-treated (Supplementary Fig. 5b, c). Time-lapse experiments indicated that Ci redistributes from LDCs to diffuse through the cytoplasm within minutes, concomitantly with DNA degradation (Supplementary Movies 6 and 7).

These data are consistent with LDCs growth through addition of diffusing Ci-DnaK and ATP-dependent disassembly leading to Ci release, which either re-integrates into the DnaK cycle or undergoes proteolysis. In addition, we show that the nucleoid acts as a physical barrier confining LDCs to the bacterial pole and that upon DNA degradation, LDCs reorganize to diffuse within the nucleoid-free space.

**LDCs exchange with the mobile fraction of Ci and DnaK.** LDCs form a dynamic matrix of selective permeability, likely resulting from constant association/dissociation of DnaK-Ci complexes. To gain insight into the association/dissociation rates of these complexes in vivo, fluorescence recovery after photobleaching (FRAP) experiments were performed. For these experiments, the LDC-

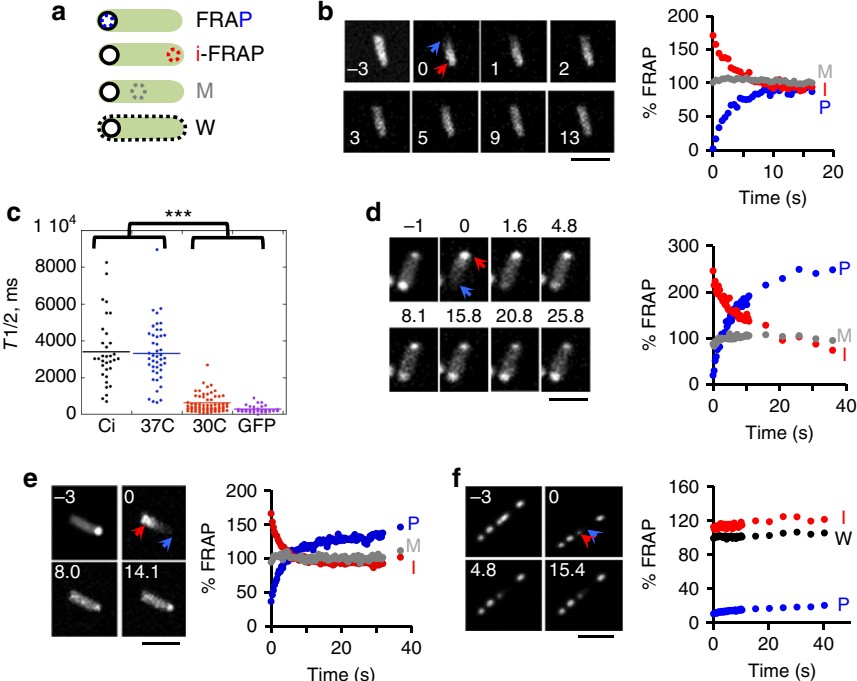

Fig. 5 FRAP analysis reveals Ci and DnaK-RFP exchanges between LDCs at opposite poles. **a** Scheme of the analytical procedure. Empty circle: photo-bleached area. Dotted area: regions subjected to FRAP analysis, corresponding to: the FRAP area (P, blue), the opposite pole (I, red), the mid- (M, gray) or whole bacterial body (W, black). **b**, **d**–**f** FRAP analysis. Left panels: representative time-lapse images of bacteria subjected to FRAP analysis. The time after bleach is indicated in seconds. Scale bar = 5 μm. Right: representative fluorescence recovery kinetics of area corresponding to FRAP area (P, blue), opposite pole (I, red) and mid-bacterial body (M, gray) analysis. The values are expressed as relative percent of the residual average intensity in whole bacterial body (W, black) after photobleaching. **c** Average half-recovery time calculated from single exponential fits for at least 30 measurements in three independent experiments. Asterisks indicate statistical significance determined by Dunn's test: $p < 3.8 \times 10^{-12}$. **b**, **d** MC4100/pDnaK-RFP grown at 30 °C (**b**) or 37 °C (**d**). **e** MC4100/pCi. **f** *dnaKJ*/pCi. The red circles correspond to analysis in a Ci patch adjacent to the photobleached area

containing pole was photobleached and fluorescence recovery analyzed in a 500-nm-diameter circular area selected in the photobleached region, at mid-cell or at the non-photobleached pole (Fig. 5a).

We performed FRAP experiments on bacteria expressing DnaK-RFP alone (MC4100/pDnaK-RFP) following a 60 min induction. As expected, bacteria grown at 30°C showed diffuse DnaK-RFP localization (Fig. 5b). Photobleaching at one pole was rapidly followed by DnaK-RFP diffusion leading to homogeneous protein distribution. FRAP curves were fitted with single exponential curves with an average half-time recovery ($T_{1/2}$) of 603 ± SEM 296 ms (Fig. 5b, c), consistent with free or only weakly restrained diffusion. As expected, FRAP curves from control experiments in wild-type bacteria expressing GFP were fitted by a single exponential equation model with a $T_{1/2}$ of 281 ± 133 ms, consistent with free diffusion (Fig. 5c). As expected, DnaK was bipolar in bacteria grown at 37 °C[25] (Fig. 5d). FRAP curves of bacteria grown at 37°C showed an exponential recovery with an average $T_{1/2}$ of 3.3 s ± 1.6 s, indicating that DnaK-RFP was not freely diffusible (Fig. 5c, d). Inverse-FRAP (I-FRAP) analysis of the non-photobleached pole showed a decrease in DnaK-RFP fluorescence, inversely mirroring the FRAP curve. These results indicate that DnaK-RFP exchanged between poles (Fig. 5d). The decrease in I-FRAP mirrored the recovery in the photo-bleached region, which suggests that the dissociation/association rates of DnaK-RFP from LDCs between poles determine the rate of fluorescence recovery.

FRAP curves for polar Ci were fitted by a single exponential equation model, with an average $T_{1/2}$ of 3.4 ± 0.2 s, similar to that observed for polar DnaK-RFP (Fig. 5c, e). I-FRAP analysis indicated that fluorescence recovery was associated with a

corresponding decrease at the non-photobleached pole with similar rates in 89.5% of events (Fig. 5e and Supplementary Fig. 8). In contrast, altered fluorescence intensity in the mid-region was only detected in 10.5% of events (Fig. 5e and Supplementary Fig. 7), suggesting that despite its unipolar accumulation, Ci exchanges occur preferentially between poles. Recovery dynamics changed following Colicin E2 treatment with no fluorescence recovery in 38.7% of cells and in 41.9% of cells fluorescence recovery occurred from both the non-photobleached pole and mid-body region (Supplementary Fig. 8). This suggests that nucleoid density plays a role in determining Ci pole-to-pole exchanges (Supplementary Fig. 8). FRAP experiments on large Ci patches in Δ*dnaKJ* showed that Ci was virtually non-diffusible with no observable recovery (Fig. 5f). The absence of Ci diffusion was independent of region localization, indicating that in Δ*dnaKJ*, Ci forms non-diffusible aggregates that exclude the nucleoid.

Together, these results indicate that Ci and DnaK in LDCs form dynamic structures that exchange between poles. The similar recovery halftimes for Ci and DnaK-RFP suggest that DnaK binding to Ci drives the incorporation of complexes in LDCs and that dissociation of these complexes from LDCs is rate limiting during the exponential recovery phase.

**DnaK and DnaJ assist LDC exchange with immature Ci.** In 51% of cases, extended FRAP kinetics showed a plateau that lasted over 80 s after exponential recovery of Ci (Fig. 6a). Interestingly, in 48.6% of cases (15/31 events), exponential recovery was followed by a linear increase in fluorescence at the pole that plateaued after extended kinetics (Fig. 6b). When pronounced, the linear increase in fluorescence was associated with a steady

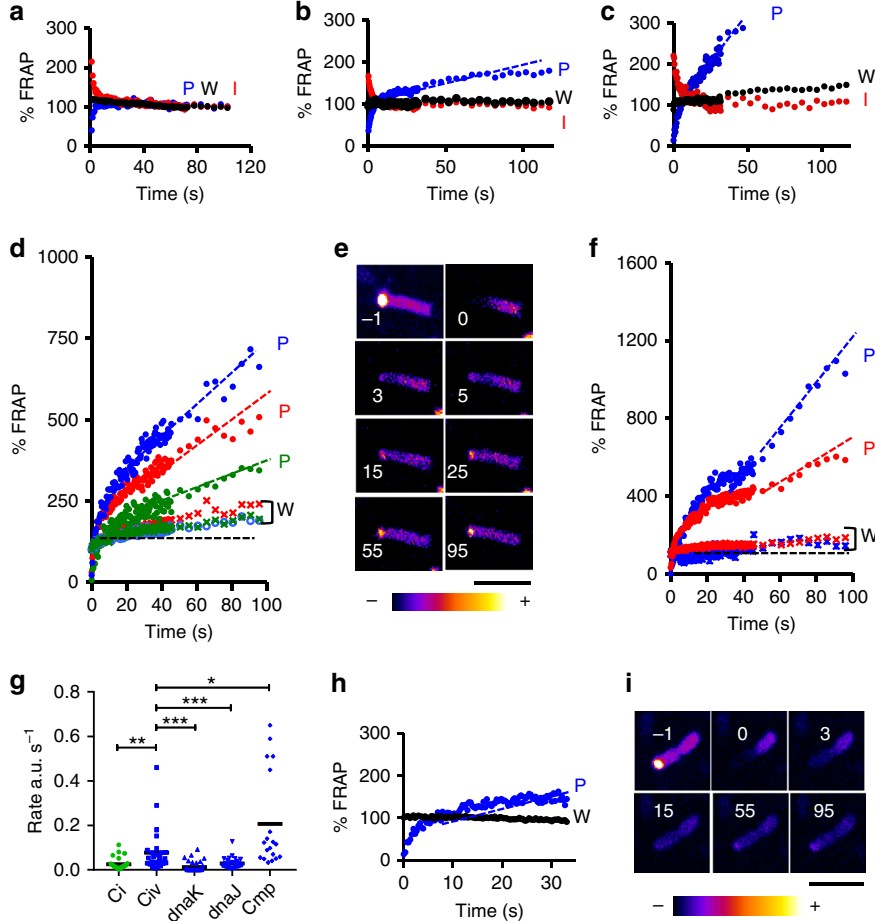

**Fig. 6** LDC growth is associated with a steady rate of DnaKJ-dependent Ci folding. FRAP analysis of MC4100/ pCi (**a**–**c**), MC4100/pCi$_v$ (**d**–**f**), or *dnaJ*/pCi$_v$ (**h**, **i**). Representative fluorescence recovery kinetics of area corresponding to FRAP (P), opposite pole (I) and whole bacterial body (W) (**a**–**d**, **h**). The values are expressed as relative percent of the residual average intensity in W following photobleaching. Linear fits corresponding to the second components are shown as dotted lines. **d**, **f** The different colors correspond to single bacteria with the corresponding P (circles) or W (dotted line) analysis. **e**, **i** Representative time-lapse images with the indicated pseudocolor code of bacteria subjected to FRAP analysis. The time after photobleach is indicated in seconds. Scale bar = 5 μm. **g** Average rate of Ci or Ci$_v$ fluorescence increase determined from linear fits (30 bacteria, $N = 3$). Ci: MC4100/pCi; Civ: MC4100/pCi$_v$; dnaK: *dnaK*/pCi$_v$; *dnaJ*: dnaJ/pCi$_v$; Cmp: MC4100/pCi$_v$ treated with chloramphenicol. Number of asterisks indicate statistical significance determined by ANOVA, *: $p < 0.05$; **: $p < 0.01$; ***: $p < 0.005$. **a** Exponential recovery of Ci fluorescence followed by a plateau phase, as illustrated in **b**, **c**, exponential recovery may be immediately followed by a linear recovery phase associated with an increase of the total bacterial fluorescence. The I-FRAP analysis shows the clear distinct exponential and linear components

increase in Ci fluorescence throughout the bacterial body (Fig. 6c). During the linear increase phase, no inverse correlation between the linear fits in the FRAP and I-FRAP regions was detected, implicating exchanges with mobile fractions not limited to the opposite pole (Fig. 6b, c). This global increase in Ci fluorescence was not observed in a *dnaKJ* mutant, consistent with a role for DnaKJ in Ci folding (Fig. 5f), indicating that the global linear increase in Ci fluorescence might result from the incorporation of immature Ci molecules lagging behind neosynthesis. In this case, the rates of linear increase in Ci fluorescence would be limited by the rate of GFP folding and maturation. To test this, we used a Ci$_v$ construct in which IpaC is fused to the YFP variant Venus that matures 5 times more rapidly than eGFP[36]. A linear increase in Ci$_v$ fluorescence was observed in 96% (27/28) of cases and, as expected this occurs faster than in cells with Ci (Fig. 6d–f). In many cases, the linear increase of Ci$_v$ levels in the cells was concomitant with pole-to-pole exchange during initial photobleach recovery. However, linear increases occasionally followed an initial plateau of the mono-exponential recovery component, consistent with two distinct steps in LDC formation (Fig. 6f). The

rates of Ci$_v$ fluorescence increase extrapolated from linear fits were plotted (Fig. 6g). As expected, this analysis showed that Ci$_v$ folds faster than Ci, with an average rate of $79 \pm \mathrm{SEM}\ 17\ \mathrm{s}^{-1}$ and $26 \pm 4$, respectively. Additionally, in 69.4% (24/3) of cases no detectable Ci$_v$ recovery occurred in the *dnaK* mutant and, when there was recovery, the average rate of fluorescence increase was slower averaging $8.2 \pm 0.3\ \mathrm{s}^{-1}$, indicating that like Ci, Ci$_v$ requires DnaK for folding and maturation (Fig. 6g). The fluorescence recovery defect in the *dnaJ* mutant was less pronounced with no detectable recovery in 37.5% (15/40) of bacteria and when there was recovery, the average rate was $19.3 \pm 0.4\ \mathrm{s}^{-1}$ (Fig. 6g). In addition, in Δ*dnaK* and Δ*dnaJ*, no increase in global Ci$_v$ bacterial fluorescence was detected, even when slow recovery was observed at LDCs, as illustrated for the *dnaJ* mutant (Fig. 6h, i). Control experiments inhibiting protein translation with chloramphenicol indicate that the linear increase in fluorescence was not due to de novo Ci$_v$ synthesis (Fig. 6g). These results are consistent with DnaKJ being important for Ci$_v$ integration into LDCs at the bacterial pole, and promoting folding of non-fluorescent immature Ci$_v$.

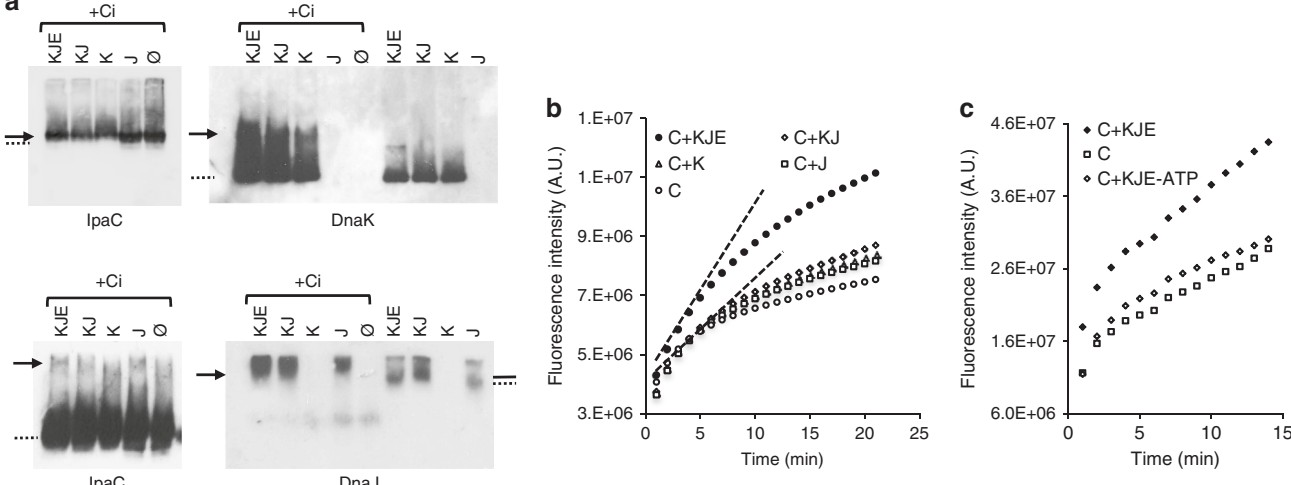

**Fig. 7** Ci folding is assisted by DnaKJE. **a** Native gel analysis of DnaKJ-Ci complexes. Indicated purified proteins were mixed and heated for 10 min at 45 °C prior to native gel analysis and western-blotting with antibody directed against the indicated protein. O with a slash indicates Ci alone. Left: anti-IpaC western blot with short (top) and long (bottom) exposure time. Arrow: Ci-induced shift; solid line: DnaK-induced shift; dotted line: native major band. **b**, **c** Refolding assay of acid-denatured Ci in the presence of the indicated chaperone. Data points represent the mean value from three independent experiments. For each value, the standard deviation is less than 10%. Sample mix: K: DnaK; J: DnaJ; E: GrpE. Ci, DnaK, DnaJ, and GrpE were incubated at the final concentration of 3, 3, 1, and 0.6 μM, respectively. **b** The dotted line represents the maximal slope used to determine the initial folding rate

**The DnaKJE machinery assists Ci folding**. To determine whether Ci was targeted by DnaKJ, protein complexes were visualized using native gel assays[23]. Ci was incubated with DnaK, DnaJ and GrpE and following native gel electrophoresis and membrane transfer, filter replicas were analyzed by western-blotting.

Incubation of purified DnaK and Ci led to a band shift that co-migrated with a Ci band shift (Fig. 7a), suggesting complex formation. This shift was accompanied by smearing of the DnaK signal, particularly pronounced in the presence of DnaJ and GrpE (Fig. 7a). We hypothesize that this smear may result from DnaJ stimulating DnaK ATPase activity generating several DnaK conformers. Probably because of its basic isoelectric point DnaJ-containing species showed reduced migration (Fig. 7a). As expected, DnaJ migration was altered when incubated with DnaK, consistent with the formation of DnaKJ complex (Fig. 7a). This shift was reverted in the presence of GrpE, consistent with the GrpE-mediated stimulation of ATP-ADP exchange and with the resetting of the DnaK chaperone cycle (Fig. 7a). Ci also induced a shift in DnaJ migration that appeared to differ from the DnaK-mediated shift (Fig. 7a). The Ci-mediated shift in DnaJ also corresponded to a shift in Ci migration. Such Ci-mediated DnaJ shift was also observed in the presence of DnaK and GrpE (Fig. 7a). These results indicate that Ci can be targeted by DnaKJ and suggests the DnaKJE machinery assists Ci folding.

We performed modified small-scale in vitro Ci folding assays[37]. After rapid acid-denaturation of Ci in the absence of chaperones, the pH was raised to neutral in the presence of DnaK, DnaJ, and GrpE. Samples were spotted onto parafilm and fluorescence emission was monitored with a CCD camera imager. In the presence of DnaK and DnaJ there was a modest increase in the Ci-refolding rate, whereas in the presence of the DnaKJE machinery and ATP the Ci refolding rate was dramatically increased (Fig. 7b). As expected, Ci-refolding by DnaKJE was ATP-dependent (Fig. 7c).

These results are consistent with DnaK–Ci complex formation and a role for DnaKJE in Ci folding.

**GrpE controls LDC formation**. GrpE stimulates ADP/ATP exchange on DnaK to facilitate substrate release, regenerating a pool of DnaK-ATP available to bind new substrates[28,29,38]. Therefore, we expected GrpE to promote dissociation of ADP–DnaK–Ci complexes from LDCs. To test the effect of *grpE* expression on LDCs, plasmids encoding IPTG-inducible Ci and arabinose-inducible GrpE were introduced into MC4100. Following Ci induction, Ci expression was inhibited and *grpE*-production was induced with arabinose. In the 30 min following arabinose induction, GrpE levels were seven times higher in the MC4100/pSUCi/pBAD22-GrpE overproducing strain compared to MC4100/pSUCi/pBAD22, and GrpE-levels remained constant over 1 and 2 h incubations (Fig. 8a). Ci levels decreased over time with a 25% and 65% reduction of Ci in the GrpE-overproducing and control cells, respectively (Fig. 8a). Consistently, when samples were analyzed by fluorescence microscopy, following IPTG-induction, all bacteria contained unipolar Ci (Fig. 8b, c). After 2 h incubation in the presence of arabinose, MC4100/pSUCi/pBAD22 showed an average reduction of 49.0 ± 2.9% and a 76.4 ± 1% in fluorescent bacteria and bacteria with polar Ci localization, respectively. In comparison, MC4100/pSUCi/pBAD22-GrpE showed a reduction of 24.7 ± 1.8% and 35.1 ± 6.0% in fluorescent bacteria and bacteria with polar Ci localization, respectively. These findings indicate that GrpE regulates LDC formation and suggest that GrpE delays Ci turnover by stimulating Ci association to and dissociation from LDCs.

## Discussion

Maintaining a cytoplasmic pool of proteins available for rapid secretion presents a challenge for pathogenic bacteria, such as T3SS-expressing *Shigella*. Dedicated T3SS chaperones prevent premature oligomerization and aggregation of type III substrates in the bacterial cytoplasm[39]. We show that in addition to its cognate chaperone IpgC, the general chaperone DnaK is critical for preventing IpaC aggregation. We provide evidence that DnaK–IpaC complexes, organized as LDCs, are polarized via nucleoid occlusion.

DnaK mediates the reversible association of Ci in LDCs found at the bacterial pole. FRAP experiments demonstrate that, unlike aggregates, Ci undergoes dynamic exchange with a mobile fraction, predominantly located at the opposite pole, suggesting

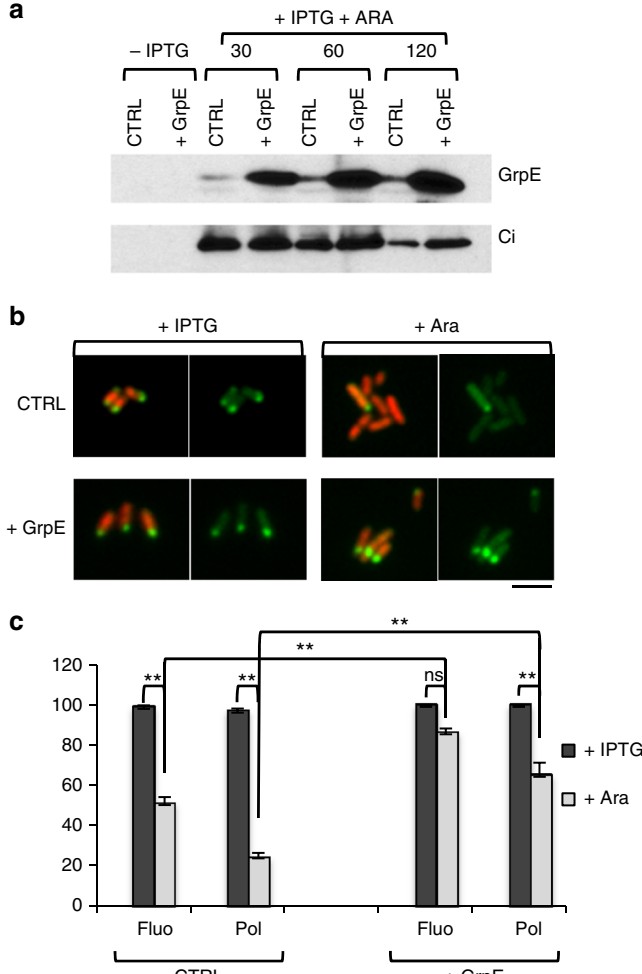

**Fig. 8** GrpE controls LDC dynamics. Following induction of Ci expression (IPTG), bacterial strains were incubated in the presence of arabinose ( + Ara) for 2 h. CTRL: MC4100/pSUCi/pBAD22; + GrpE: MC4100/pSUCi/pBAD22-GrpE. **a** Bacterial lysates were analyzed by western-blotting with the indicated antibody. **b, c** Samples were fixed and processed for fluorescence microscopy analysis. **b** Representative micrographs. Red: DAPI staining; Green: Ci fluorescence. Scale bar = 5 μm. **c** Percent (mean ± SEM) of bacteria ( >150 bacteria, $N = 2$) with fluorescent signal (Fluo), and bacteria with polar Ci signal (Pol). Statistical significance was determined by ANOVA with Tukey's post hoc analysis (**, $P \leq 0.01$; ns, not significant $P > 0.05$)

DnaK–Ci complexes are present at both poles, but asymmetrically distributed. Additionally, the average Ci fluorescence at the opposite pole is equal or lower than at mid-cell. Therefore, it is likely that two pools of Ci exist in the mobile fraction, yet based on FRAP analyses Ci-DnaK at the opposite pole exchanges more readily than the cytosolic pool present in the nucleoid. The latter likely corresponds to neosynthesized immature Ci complexed with DnaK. This is consistent with polysomes binding co-transcriptionally to neosynthesized mRNA in the nucleoid and then accumulating at the bacterial poles[40].

LDCs appear to act as DnaK sequestering hubs, with up to 80% of the total chaperone pool sequestered in LDCs. Slow incorporation of Ci into LDCs observed during the second recovery in our FRAP experiments suggests ATP-DnaK is the rate-limiting factor in LDC biogenesis. LDC-forming kinetics are influenced by GrpE-mediated DnaK recycling from LDCs and DnaKJ-mediated loading of Ci into LDCs (Fig. 9b). Due to the reversible and dynamic nature of LDCs, GrpE activity may potentiate LDC formation by increasing the pool of free ATP-DnaK, boosting the activity of the DnaKJE machinery, thereby increasing the resident time of LDCs. Conversely, in the absence of GrpE, DnaK recycling from LDC stalls, slowing LDC growth.

DnaK and DnaJ are involved in polar accumulation of large aggregates that are eliminated following asymmetric inheritance in daughter cells[25]. In contrast, we present evidence that transient ADP-bound DnaK and IpaC interactions are involved in LDC polarization. In support, DnaK overexpression shifts the predominantly unipolar Ci-containing LDC to a predominantly bipolar localization, which can be explained through capture-diffusion mechanisms. At physiological DnaK levels, DnaK–Ci complexes are confined to one pole mostly through Ci–Ci interactions. LDC birth and growth occur through capture of neosynthesized Ci bound to DnaK via IpaC–IpaC interactions. Increasing the DnaK:Ci ratio results in Ci localization being driven by the bipolar DnaK distribution via DnaK–DnaK interactions. Higher order ADP-DnaK homo-oligomers were convincingly documented in independent studies[23,24,41]; yet, few are detected in vivo, consistent with labile interactions[41]. Oligomerization may also occur through a domain of the client protein in complex with DnaK. IpaC-sequence analysis identified DnaK-binding motifs consisting of a hydrophobic region flanked by basic residues[29,42]. IpaC constructs lacking this motif localized diffusely and were poorly secreted by the T3SS[11], in line with the role for DnaK described here. One predicted DnaK-binding site overlaps with the IpaC oligomerization domain, which indicates that diffusion-capture of DnaK–IpaC complexes might not occur through this region. Further analysis is needed to investigate the interactions leading to DnaK oligomer formation in vivo, a difficult task because of the large repertoire of client proteins and DnaK conformers.

We propose the following model: ATP-DnaK binds co-translationally to nascent IpaC, assisted by DnaJ (Fig. 9a). ATP hydrolysis leads to stabilization of ADP–DnaK–Ci complexes that self-associate through the IpaC oligomerization domain or a DnaK subdomain interface specific to the ADP-bound form (Fig. 9a). While reversible, these self-interactions form LDCs that are sequestered to the pole through nucleoid occlusion. At steady state, LDCs constantly exchange DnaK–Ci complexes between opposite poles (Fig. 9a) and grow by incorporating neosynthesized Ci (Fig. 9a). An unknown quality control machinery controls Ci-levels through proteolysis after it dissociates from LDCs (Fig. 9b). GrpE-mediated acceleration of DnaK-ADP/ATP nucleotide exchange leads to the release of Ci and ATP-DnaK that becomes free to interact with neosynthesized substrates (Fig. 9b). These results suggest that transient interactions within LDCs drive polar substrate localization, probably through transient capture-diffusion while being subjected to nucleoid occlusion. LDCs may serve as a reservoir of substrates that need to be mobilized rapidly for secretion during activation of the T3SS.

The role of the DnaKJE machinery might explain why IpaC polarization is required for efficient secretion. In other secretion systems, cognate chaperones of specific secretion systems and main chaperones, such as DnaK, have been implicated in the export of substrates or the correct folding of membrane proteins[43]. Previous studies showed that DnaK is required for epithelial cell invasion by *Salmonella enterica* and type III secretion[44]. Our inability to introduce a *dnaK* deletion in *Shigella* by transduction suggests that DnaK is essential and is consistent with the lethal aggregation observed here. Type III substrates that misfold in the absence of DnaK may jam T3SS and prevent secretion. Indeed, the T3SS entry gate and needle have an estimated diameter of 2–3 nm, requiring the type III ATPase to unfold type III substrates prior to secretion[11,45]. In *Shigella*, IpaC

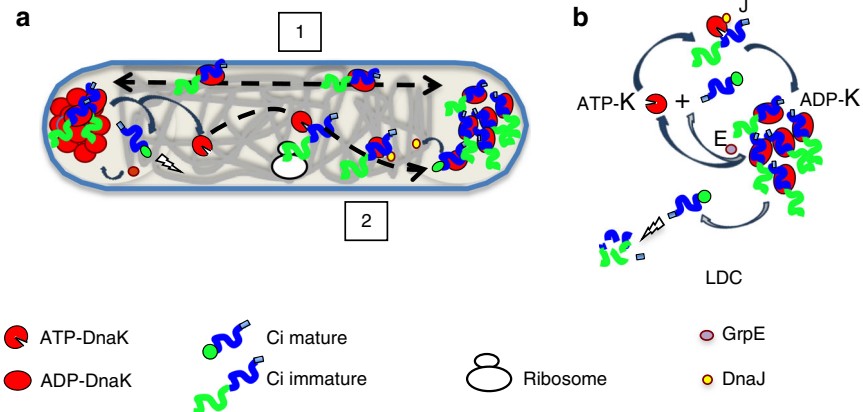

**Fig. 9** Model for LDC formation and dynamics. **a** LDCs act as hubs for mobilizable IpaC. Oligomeric ADP-DnaK:Ci complexes are confined to the bacterial pole as part of LDCs through reversible intermolecular interactions, and ADP-DnaK:Ci complexes exchange between opposite poles (1). These transient interactions may be mediated by the self-oligomerization domain of the client protein Ci, or via oligomerization domains on ADP-DnaK (1). ATP-DnaK binds to neosynthesized Ci. DnaJ may assist DnaK in IpaC recognition and stimulate ATP hydrolysis of DnaK leading to the formation of stable ADP-DnaK: Ci that can be integrated into LDCs (2). Under steady state conditions, this process is slow and limited by the available pool of ATP-DnaK. This process can be modulated by increasing GrpE-levels to accelerate the conversion rate of ADP-DnaK to ATP-DnaK, which will speed up the dissociation rate of DnaK from LDCs to increase the pool of free ATP-DnaK. **b** By sequestering ADP-DnaK: Ci complexes, LDCs limit the pool of free ATP-DnaK. Consequently, the activity of GrpE is critical for maintaining the available pool of ATP-DnaK and for loading of ATP-DnaK:Ci complexes into LDCs. Ci, dissociated from LDCs by GrpE are subjected to proteolytic degradation by the quality control machinery

variants that do not polarize are not effectively secreted[11], suggesting that DnaK may play a role in maintaining type III substrates in a secretion-competent conformation. Consistently, oligomeric ADP-bound DnaK was shown to have "holdase" activity[23], which in LDCs, may help lowering the energy required for substrate translocation through the T3SS. More work is needed to characterize functional interactions between the DnaKJ machinery and the type III ATPase and chaperones. LDCs provide *E. coli* and related bacteria with a means to establish a pool of ready-to-go substrates, critical for time-sensitive processes. Specifically, within seconds following cell contact type III secretion substrates are injected into host cells from the bacterial pole, the spatio-temporal aspects of secretion likely being important for signalling leading to bacterial invasion[11,46].

## Methods

**Bacterial strains and plasmids.** Wild-type *E. coli* strain MC4100 F- ([araD139]B/r Δ(argF-lac)169* λ- e14- *flhD5301* Δ(*fruK-yeiR*)725 (*fruA25*)‡ *relA1 rpsL150*(Str[R]) *rbsR22* Δ(*fimB-fimE*)632(::IS1) *deoC1*), *cbpA* and *djlA*, as well as the triple *cbpA*, *djlA*, *dnaJ* isogenic mutants were used in this study[47]. The *dnaK* and *dnaJ* mutants are MC4100 isogenic strains carrying the Δ*dnaKJ*::Kan[R] *thr*::Tn*10* and Δ*dnaJ*::Tn*10*-42 (Tet[R]) mutations, respectively[48,49]. Bacteria were grown in LB medium at 37 °C or 30 °C containing 0.4% glucose when indicated. The strains were transformed with plasmids pCi$_v$ or pSUCi, expressing full-length IpaC fused to Venus or GFP, respectively[47] and/or the P17A plasmid encoding the IpgC chaperone[50]. Exponentially grown bacteria were induced with 100 nM IPTG for 60 min. When needed, antibiotics were added at the following final concentration: ampicillin, 50 μg/mL; kanamycin, 50 μg/mL; tetracyclin, 5 μg/mL; zeocin, 25 μg/mL. pDnaK-RFP was generated by PCR cloning *dnaK* into the zero Blunt TOPO vector (Invitrogen) using the following primers: 5′- GGAGACGTTTAGATGGGTAAAATAATTGG-3′ and 5′- GACTTAAACTTCTTCAGTTTCTGTTTTTT-3′ to generate pCR-Blunt-DnaK. The RFP ORF was amplified by protein using the following primers: 5′- GAATCCTCGAGATGGCCTCCTCCGAGGA-3′ and 5′- ATTCTCTA-GATTAGGCGCCGTGGGAGT-3′, and cloned into the XhoI and XbaI sites of pCR-Blunt-DnaK-RFP to generate pDnaK-RFP. The T199A mutation was introduced in pDnaK-RFP using the QuikChange® Site-Directed Mutagenesis Kit using the following primers: 5′- GACCTGGGTGGTGGTGCCTTCGATATTTCTATTATCG-3′ and 5′- CGATAATAGAAATATCGAAGGCACCACCACCCAGGTC-3′ to generate pDnaK-T199A-RFP. pCiΔU-His was obtained by PCR cloning the CiΔU ORF from pCiΔU into the Champion pET Directional TOPO® vector (Invitrogen)[11] using the following primers: 5′- CACCATGGAAATTCAAAACA-CAAAACCAAC-3′ and 5′-AGCTCGAATGTTACCAGCAATC-3′. All constructs were verified by DNA sequencing.

**Protein immunoprecipitation.** CiΔU-His and associated proteins were purified using Talon resin following cross-linking (Clontech). Briefly, an exponentially grown culture of WT/pCiΔU-His was induced with IPTG, and bacteria were harvested by centrifugation. All subsequent steps were performed at 4 °C unless otherwise stated. Bacteria were washed in 50 mM HEPES pH 7.4, 100 mM NaCl containing CompleteTM, (Roche Pharmaceuticals). When indicated, samples were resuspended in the same buffer containing 0.1% formaldehyde. Samples were incubated for 20 min at 21 °C. Tris-HCl was added at 100 mM final concentration and bacteria were pelleted by centrifugation. Samples were resuspended in the same buffer containing 0.1% Triton X-100 (lysis buffer), and bacteria were lysed using a cell disruptor (Branson Sonifier 250) at maximal output for 10 min at 30 % duty cycle. Lysates were clarified by centrifugation for 15 min at 17 K using an Optima L (Beckamn Coulter) centrifuge. The clear supernatant was incubated for 90 min with the Talon resin pre-equilibrated in lysis buffer. Samples were eluted in lysis buffer containing 200 mM imidazole. Samples were boiled for 10 min and analyzed by SDS-PAGE and Coomassie staining and anti-IpaC western blotting. CiΔU-His-containing fractions were dialyzed against 0.1% Triton X-100, 25 mM HEPES pH 7.5, 100 mM NaCl, aliquots were flash-frozen and stored at -80 °C.

**Fluorescence microscopy.** The bacterial strains were grown in LB supplemented with appropriate antibiotics until OD$_{600 \text{ nm}}$ = 0.1. Following a 1-h induction of Ci$_v$ with 100 nM IPTG, bacteria were washed by centrifugation in PBS, and immobilized onto coverslips coated with poly-L-lysine. Samples were fixed in 3.7% paraformaldehyde for 20 min and processed for fluorescence labeling[11]. Time-lapse microscopy of Ci$_v$-expressing bacteria was performed as follows[51]. After IPTG induction, bacteria were diluted to OD 0.04 and 2 μl of the diluted cells were spotted onto 5 mm² LB-agarose (2%) pads. Liquid was allowed to absorb into the pads for 5 min in a biosafety cabinet and then pads were inverted onto Ibidi-chambers that were sealed with vacuum grease to prevent drying. Samples were analyzed on a heated stage (37 °C) using a LEICA DMRIBe inverted microscope equipped with Coolsnap HQ2 camera and LED source lights (Roper Instruments), driven by the Metamorph 7.7 software (Universal imaging). Acquisition was performed using a 63/1.25 HCX PL APO objective and fluorescence band pass filters (excitation 480 ± 20 nm, emission 527 ± 30 nm). Alternatively, samples were analyzed by spinning disk confocal fluorescence microscopy using a Nikon Elipse Ti inverted microscope and a equipped with a 100 × objective (NA 1.4), a CSU-X1 spinning disk confocal head (Yokogawa), an Evolve camera (Roper Scientific Instruments and controlled by the Metamorph 7.7 software. Within experiment series, identical acquisition conditions were used, controlling that no image saturation occurred for quantitative analysis.

**Image analysis.** Cell outlines and position of fluorescent foci relative to the bacterial midline were detected using the MicrobeJ plugin for ImageJ[52]; outlines and foci were saved as regions of interest and kept as overlays on the microscopy images. When DAPI and Ci foci were detected together, MicrobeJ standard normalization was used to set the peak fluorescence intensity of a given field to 1. For Ci and Ci$_v$ experiments, peaks in fluorescence intensity were determined by calculating the area under the curve (GraphPad Prism), peaks in fluorescence were

 ARTICLE

then plotted relative to their position in bacterial cells at each time-point. When indicated, fluorescence was normalized by subtracting the average fluorescence intensity of a non-relevant area devoid of bacteria from the average fluorescence intensity of the Ci polar region. Tracking videos were generated using the Trackmate plugin for ImageJ and exported using Icy software[53].

**FRAP analysis.** Bacteria were mounted in an observation chamber on an inverted spinning disk confocal microscope (Nikon Eclipse Ti) analyzed with a 100× objective lens, using an EM-CCD camera Evolve equipped with a FRAP module and driven by the Metamorph software (Roper Scientific Instruments). For each sample, bleaching and image acquisition were performed using identical conditions, with bleach area size of $1 \pm SEM\ 0.2\ \mu m^2$. Bleaching was performed using a 470 nm excitation laser for 80 ms until fluorescence was reduced to background levels. Image acquisition was performed every 300 ms, with 200 frames during pre-bleach and 2000 frames to monitor fluorescence recovery. Relative fluorescence intensity was determined as FRAP % = $(F_t /C_t) / (F_0/C_0) \times 100$, where $F_0$ and $C_0$ represent the average intensity region and the total bacterial body, respectively, before bleaching, and $F_t$ and $C_t$ represent the average intensity at time $t$ after bleaching for the same corresponding regions. Fluorescence recovery curves corresponding to the normalized intensity in the bleached area averaged from several measurements showed good quality fit with a single-component diffusion.

**Native gel analysis of protein complexes.** CiΔU-His was purified using nickel affinity chromatography following the manufacturer's instructions (Talon resin, BD Biosciences). DnaK, DnaJ and GrpE were purchased from StressGen Biotechnologies Corp. Proteins were mixed at 4°C in refolding buffer containing 25 mM HEPES-KOH pH 7.5, 100 mM potassium acetate, 10 mM magnesium acetate, 2 mM dithiothreitol (DTT). When indicated, ATP was added at a final concentration of 3 mM. Proteins were used at the following concentrations: CiΔU-His: 3 μM; DnaK: 3 μM; DnaJ: 670 nM; GrpE: 330 nM. After incubation for 30 min at 45 °C, CiΔU-His was diluted in refolding buffer in the presence of the indicated chaperones. Samples were incubated for 15 min at 21 °C, and separated onto a 7.5% polyacrylamide native gel[54]. Filter replicas were subjected to anti-IpaC or anti-DnaK western-blotting.

**Antibodies.** Mouse monoclonal antibodies used in western blot experiments were used at a dilution of 1/1000 for anti-RpoB (Neoclone Cat W0023), anti-DnaK and anti-GroEL[55]. The rabbit polyclonal antibodies were used at dilutions of 1/1000 anti-GrpE[56] and 1/3000 anti-DnaJ[56,57], 1/2000 anti-MBP[58] and 1/10,000 anti-IpaC[11,59,60].

**In vitro protein refolding.** CiΔU-His folding was measured using GFP fluorescence detection[61]. CiΔU-His at a final concentration of 20 μM was acid-denatured with 33 mM HCl for 10 min at 21 °C. The pH was raised to 7.5 with an equal volume of NaOH, and CiΔU-His was aliquoted into tubes containing chaperones (DnaK, DnaJ and GrpE at 3, 1, and 0.6 μM, respectively) in concentrated buffer (final concentration 1 mM DTT, 0.3 mM EDTA (ethylenediaminetetraacetic acid), 50 mM Tris pH 7.5). Drops (5 μl) were immediately spotted onto parafilm, placed in the humidified chamber of a Fujifilm LAS4000 imager equipped with a CCD camera. Images were acquired every 60 s in GFP detection mode using the 510DF10 filter.

**Statistical analyses.** For all samples, significance relative to control samples was tested using an appropriate statistical test. When possible, we chose to use sample sizes composed of at least 30 bacteria. Normality and $F$ tests were performed using Prism software; when comparing two groups an unpaired Student's $t$ test with unequal variance was used and when performing multiple comparisons ANOVA's or non-parametric tests with appropriate post hoc tests were used.

**Data availability.** Uncropped gels and blots are available in the Supplementary Information section (Supplementary Figs. 9 and 10). For data distribution, graphs with overlaid individual data points are available (Supplementary Fig. 11). Data supporting the findings of this study are available from the corresponding authors upon request.

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

## Acknowledgements

The authors are grateful to members of the team "Intercellular Communications and Microbial Infections" of the CIRB for their constant support. We thank Drs. Debabrata Raychaudhuri and Andrew Wright (Tufts University School of Medicine, Boston) for helpful discussion and providing us with the *E. coli* K12 strain producing Colicin E2. We thank Jérémie Teillon from the CIRB imaging facility for technical help, as well as, Daniel-Isui Aguilar-Salvador for help with Icy software. This work was funded by the Inserm, the CNRS and the Collège de France, as well as grants from the Labex Memolife "Microlife" and the PSL Idex "Shigaforce". C.C. is the recipient of an MERT grant attributed by the Université Paris-Diderot, Ecole Doctorale: B3MI. J.L.T. is the recipient of an NSERC postdoctoral fellowship.

## Author contributions

C.C., J.L.T., and G.T.V.N. conceived, performed and analyzed the experiments and wrote the manuscript. P.G. and O.L.F. planned and analyzed experiments.

## Additional information

**Competing interests:** The authors declare no competing interests.

