## [Peer Review File · Nature Communications]

Reviewers' comments:

Reviewer #1 (Remarks to the Author):

This manuscript reports work done to determine the role of DnaK in polar targeting and localisation of the TTSS protein IpaC. DnaK was initially identified through a cross-linking, pull-down and proteomics based approach. Subsequent experiments used expression of fluorescent protein (FP) tagged IpaC constructs, E. coli K12 mutants, treatments that degrade the nucleoid (colicin E2, and ciprofloxacin), and FRAP was used to determine the mobility of the FP tagged IpaC under varying conditions and situations. The FRAP experiments are elegant and informative. The experimental work is technically complex and well executed. The results are of interest to researchers outside the Shigella TTSS field. The data and statistical analysis used were appropriate.

While the data is in general convincing, several aspects of the manuscript are hard to follow and lack clarity. The authors describe a model (Fig. 9) in which DnaK in concert with ancillary proteins GrpE and DnaJ maintains IpaC in soluble complexes (Large Dynamic Complexes (LDCs) which are located at one cell pole. The complexes can cycle to the opposite pole. The nucleoid excludes these complexes from the bulk of the bacterial cell. While the model accounts for the role of DnaK/DnaJ/GrpE in Ci (= IpaC) dynamics, the authors need to explain why the LDCs with IpaC locate predominantly to one pole. Why not both poles?

Minor comments

The resolution of Figure 2 is too low.

Figure 3. 1H and 2H are not described in the legend.

Figure 4. Ci+M and M+N are not described in the legend.

Figures 5 and 6. The use of letters for the panels and for labelling lines in graphs is confusing (Fig. 5 and 6, panel F and line F)

Figure 6. Panel G resolution is too low.

Figure 7A. What is "O" ?

Suppl. Movie 1. Can the author please show a control cell that has not been treated with colicin E2. The cell is elongating with time. There is nothing with which to compare this image.

The description of the fluorescent protein used to tag IpaC is confusing. The GFP and Venus tagged IpaC (Ci and Civ) are used interchangeably in the manuscript (e.g. p. 12-14, data in Fig. 7 and Fig. 8). These proteins have different properties and potentially may behave differently.

In several places inappropriate referencing is used. p.4. l. 21; p. 6. l. 23. p. 8. l. 25. P. 16. l. 5.

p. 2. l. 15. "...role in either their function..."

p. 3. l. 2. "...formation of a ..."

p. 11. l. 16. "perduring" What does this mean?

p. 23. l. 9. "25 mg/ml" seems excessively high. What [] was used?

The concentration of Colicin E2 used is not described in the legends.

Reviewer #2 (Remarks to the Author):

The study by Collet and colleagues examines the sub-cellular organization and properties of large dynamic complexes in *e. coli*. The study identifies interesting mechanistic features of these structures, demonstrating chaperone dependence, mobility properties, and involvement in folding of IpaC intermediates.

The data concerning chaperone involvement and folding mechanism strongly support the conclusions. The element of the paper that seems weakest is the issue of nucleoid occlusion. The ColE2 etc treatments are less than convincing, since the overall cellular morphologies (and hence the internal organization and physical properties) change dramatically. Hence, I think that other mechanisms regulating localization should be considered and tested.

Minor Issues:

How does ATP depletion affect the LDCs?

Which pole do the IpaC LDCs localize (old or new)?

Reviewer #3 (Remarks to the Author):

In this work, the authors sustain that DnaK (HSP70) chaperone controls unipolar localization of the Shigella IpaC type III secretion substrate by promoting the incorporation of IpaC into large and dynamic complexes (LDCs) that become restricted at the bacterial pole by nucleoid occlusion. This is an interesting topic, following several recent works on it. However, I failed to find significant novel claims. Both the phenomena (polar localization of large complexes in bacteria) and the ‘mechanisms’ that support it (DnaK, nucleoid exclusion, cell division) are well known (reported both in works cited in this manuscript, as well as in several uncited works).

Also, further evidence is required. To prove the main conclusions, the authors caused DNA degradation (section “The nucleoid restrains Ci and DnaK at the bacterial pole”). However, this likely has many side effects that could indirectly lead to failure in polar localization, rather than the decrease in nucleoid density alone. One test, already proposed in previous works, consists in tracking aggregates and determine if the kinetics is consistent with nucleoid occlusion (see e.g. (Gupta et al, 2014, Biophysical Journal 106, 1928–1937)). A complementary test is to increase cytoplasm viscosity (Parry et al (2014) Cell 156: 1–12.) which disrupts midcell exclusion by nucleoid occlusion.

Finally, I would suggest a more detailed description of the image analysis procedures, as critical results depend on them. First, a software is mentioned (what methods does it use? Is there manual correction? etc..). The second statement (which is missing some word, I believe) mentions background correction and comparison to the rest of the body (what is compared and how?).

Point by point response to reviewer comments

Reviewer #1 (Remarks to the Author):

This manuscript reports work done to determine the role of DnaK in polar targeting and localisation of the TTSS protein IpaC. DnaK was initially identified through a cross-linking, pull-down and proteomics based approach. Subsequent experiments used expression of fluorescent protein (FP) tagged IpaC constructs, E. coli K12 mutants, treatments that degrade the nucleoid (colicin E2, and ciprofloxacin), and FRAP was used to determine the mobility of the FP tagged IpaC under varying conditions and situations. The FRAP experiments are elegant and informative. The experimental work is technically complex and well executed. The results are of interest to researchers outside the Shigella TTSS field. The data and statistical analysis used were appropriate.

While the data is in general convincing, several aspects of the manuscript are hard to follow and lack clarity.

The manuscript was reworked throughout to improve clarity.

The authors describe a model (Fig. 9) in which DnaK in concert with ancillary proteins GrpE and DnaJ maintains IpaC in soluble complexes (Large Dynamic Complexes (LDCs) which are located at one cell pole. The complexes can cycle to the opposite pole. The nucleoid excludes these complexes from the bulk of the bacterial cell. While the model accounts for the role of DnaK/DnaJ/GrpE in Ci (= IpaC) dynamics, the authors need to explain why the LDCs with IpaC locate predominantly to one pole. Why not both poles?

The mechanism underlying the unipolar localization of proteins in E. coli is still poorly understood. As mentioned in the discussion p. 17, we believe that a diffusion-capture model can account for the predominantly unipolar localization of Ci, with LDCs acting as sinks for neo-synthesized Ci. In this model, the formation of a LDC depends on ADP-DnaK- Ci forming higher order structures through DnaK-DnaK and Ci-Ci interactions. In the absence of excess quantities of DnaK, the Ci-Ci interaction appears to be dominant, allowing the birth and growth of LDCs by capture of neo-synthesized DnaK bound Ci through interaction with the IpaC oligomerization domain. This is supported by both FRAP experiments and our additional data, showing birth and growth of LDCs at a single pole.

Such a diffusion-capture model also explains the preferential bi-polar localization of DnaK, expected to bind to a variety of substrates showing oligomerization / aggregation properties which will stochastically be confined at one pole or the other by nucleoid occlusion. The redistribution of Ci from uni- to bipolar upon DnaK overexpression is consistent with shifting towards DnaK-DnaK interactions predominantly controlling LDC formation, possibly because of the increased concentrations of DnaK molecules.

The text in the discussion p. 17, l. 7, has been rewritten as follows:

"These observations can be explained through simple capture-diffusion mechanisms. At physiological DnaK concentrations, DnaK-Ci complexes are confined at one pole mostly through Ci-Ci interactions. Birth and growth of

LDCs would occur through capture of neosynthesized Ci bound to DnaK via interaction with the IpaC oligomerization domain. When the DnaK: Ci ratio increases, the Ci location would be predominantly driven by the bipolar DnaK distribution, through capture of Ci-DnaK by LDCs via DnaK-DnaK interactions. "

Minor comments

The resolution of Figure 2 is too low.

This figure has been replaced with a higher resolution version

Figure 3. 1H and 2H are not described in the legend.

Panels on the right show the *dnaKJ* mutants grown for 1 hour (1H) and 2 hours (2H). The Figure legend has been changed to: "Panels on the right show *dnaKJ* mutants grown for 1 hour (1H) or 2 hours (2H)".

Figure 4. Ci+M and M+N are not described in the legend.

This figure is now Suppl. Fig. 4, all supplementary figure numbers have been changed accordingly. M denotes membrane staining with FM4-64 dye and N the nucleoid staining with DAPI, as indicated in the revised version of the legend.

Figures 5 and 6. The use of letters for the panels and for labelling lines in graphs is confusing (Fig. 5 and 6, panel F and line F)

The data line for FRAP is now labeled with the letter P and labels have been introduced in panel A, in addition to their colour code.

Figure 6. Panel G resolution is too low.

This panel has been replaced by a higher resolution image in the revised manuscript

Figure 7A. What is "O" ?

O denotes Ci alone and has been replaced in the figure by \emptyset , as indicated in the revised figure legend.

Suppl. Movie 1. Can the author please show a control cell that has not been treated with colicin E2. The cell is elongating with time. There is nothing with which to compare this image.

A control movie without Colicin E2 treatment was added as Supplementary movie 4.

The description of the fluorescent protein used to tag IpaC is confusing. The GFP and Venus tagged IpaC (Ci and Civ) are used interchangeably in the manuscript (e.g. p. 12-14, data in Fig. 7 and Fig. 8). These proteins have different properties and potentially may behave differently.

We have clarified the use of the GFP and Venus constructs in the text by expanding on the rationale (page 12, l. 8): "We reasoned that the global linear increase in Ci fluorescence was due to the incorporation of immature Ci molecules lagging behind neosynthesis. If this is true, the rates of linear increase in Ci fluorescence are expected to be limited by the slow kinetics of GFP folding and maturation. To test this, we used a Ci_v construct in which IpaC is fused to the fast-maturing variant of YFP, Venus that matures 5 times more rapidly than eGFP (Nagai, Ibata et al., 2002)."

In several places inappropriate referencing is used. p.4. l. 21; p. 6. l. 23. p. 8. l. 25. P. 16. l. 5.

The references have been reformatted and these mistakes are corrected in the revised manuscript.

p. 2. l. 15. "...role in either their function..."

The word "either" has been added in the revised version.

p. 3. l. 2. "...formation of a ...

The text was modified.

p. 11. l. 16. "perduring" What does this mean?

"perduring" has been replaced with "lasting"

p. 23. l. 9. "25 mg/ml" seems excessively high. What [] was used?

We apologize for the typographical error, "mg" was changed to "µg".

The concentration of Colicin E2 used is not described in the legends.

We realize that the methods section describing colicin E2 preparation and treatment lacked details and apologize. The text has been rewritten and moved in the Supplemental section with Suppl. Fig. 4 in the revised manuscript. The legend to Suppl. Fig. 4 has been changed accordingly.

This section now reads:

"Colicin E2 was prepared as previously described (52). Briefly, the colicinogenic strain A798 was grown to OD 600 nm = 0.1 in LB medium at 37°C in a shaking incubator. Following addition of mitomycin at a final concentration of 2 µg / ml, the bacterial culture was incubated until a final OD 600 nm = 0.5. Bacteria were pelleted by centrifugation for 15 min at 8000 x g at 4°C, and the Colicin E2-containing supernatant was transferred to a fresh tube. Chloroform was added at a final concentration of 0.01%, and the supernatant (E2 SN) was stored at 4°C. As expected, the E2 SN showed bactericidal activity when used at dilutions as low as 10⁻⁴ in LB. Bacterial DNA degradation was observed after a 15 min incubation when E2 SN was used at a dilution of 1:2, and this is the dilution used for E2SN treatments."

Reviewer #2 (Remarks to the Author):

The study by Collet and colleagues examines the sub-cellular organization and properties of large dynamic complexes in e. coli. The study identifies interesting mechanistic features of these structures, demonstrating chaperone dependence, mobility properties, and involvement in folding of IpaC intermediates.

The data concerning chaperone involvement and folding mechanism strongly support the conclusions. The element of the paper that seems weakest is the issue of nucleoid occlusion. The ColE2 etc treatments are less than convincing, since the overall cellular morphologies (and hence the internal organization and physical properties) change dramatically. Hence, I think that other mechanisms regulating localization should be

considered and tested.

Both reviewers 2 and 3 felt this was the least convincing aspect of the manuscript; we agree and performed additional experiments as suggested by the reviewers to strengthen our argument for nucleoid occlusion, please see the new Fig. 4 and our response to Reviewer 3 below.

Minor Issues:

How does ATP depletion affect the LDCs?

We have tested the effect of ATP depletion in IpaC-GFP tracking experiments by adding DNP to the bacteria (Parry et al. 2014; Fig. 4D-G). We observe stabilization of the fluorescence signal, as would be expected if we were to freeze the DnaK machinery in place. In addition, the LDCs remain trapped at the bacterial pole for as long as we could measure.

Which pole do the IpaC LDCs localize (old or new)?

Throughout our time-lapse experiments with MC4100/pCi_v, we observed that LDCs preferentially appear at the old pole (New Fig. 4I; New Suppl. video 3). We hypothesize that the pre-existing pool of IpaC at the old pole prior to cell division may cause this. This is different from what we previously visualized in *Shigella*, where we found an equal distribution between the new and old pole. While we do not have a clear explanation for this difference, it could potentially be explained by a difference in GrpE activity as suggested by the results obtained in Fig. 8 – a subject that will be pursued in further detail as part of another study.

Reviewer #3 (Remarks to the Author):

In this work, the authors sustain that DnaK (HSP70) chaperone controls unipolar localization of the Shigella IpaC type III secretion substrate by promoting the incorporation of IpaC into large and dynamic complexes (LDCs) that become restricted at the bacterial pole by nucleoid occlusion. This is an interesting topic, following several recent works on it. However, I failed to find significant novel claims. Both the phenomena (polar localization of large complexes in bacteria) and the ‘mechanisms’ that support it (DnaK, nucleoid exclusion, cell division) are well known (reported both in works cited in this manuscript, as well as in several uncited works).

Also, further evidence is required. To prove the main conclusions, the authors caused DNA degradation (section “The nucleoid restrains Ci and DnaK at the bacterial pole”). However, this likely has many side effects that could indirectly lead to failure in polar localization, rather than the decrease in nucleoid density alone. One test, already proposed in previous works, consists in tracking aggregates and determine if the kinetics is consistent with nucleoid occlusion (see e.g. (Gupta et al, 2014, Biophysical Journal 106, 1928–1937)). A complementary test is to increase cytoplasm viscosity (Parry et al (2014) Cell 156: 1–12.) which disrupts midcell exclusion by nucleoid occlusion.

To provide further evidence and clarify this point, we have now performed additional experiments that are included in the new Figure 4. As per Gupta et al. 2014, we quantified the location of complexes at the pole (Fig 4ABC) relative to the bacterial midline and DAPI staining, we also tracked LDCs during bacterial growth.

Unlike in other studies where foci movement was readily observed during bacterial growth (Gupta et al 2014, Parry et al 2014), our tracking experiments indicate that LDCs appear at the pole where they are confined. We did not observe the appearance of small fluorescent foci that move towards the poles and come together to form a complex. Instead, we observed the dynamic assembly and disassembly of LDCs at the pole. Strikingly, in the presence of the ATP-depleting compound DNP, we observed that the large polar Ci complexes were stabilized at the pole. These results are consistent with the described DNP-increasing cytoplasmic viscosity (Parry et al., 2014), with LDC dissociation requiring active metabolism and GrpE-dependent DnaK transition from its -ADP to its -ATP bound form.

Together these data suggest that LDC growth is mediated through the assembly of individual elementary Ci-DnaK complexes at the pole and that, while reversible, LDCs themselves are too large to diffuse through the nucleoid.

These new results in combination with our previous DNA-degradation data (now shown in Suppl. Fig 4) are consistent with nucleoid occlusion. These data provide evidence for a conceptual shift where nucleoid occlusion can also confine large dynamic complexes that are subjected to continuous assembly and disassembly.

Finally, I would suggest a more detailed description of the image analysis procedures, as critical results depend on them. First, a software is mentioned (what methods does it use? Is there manual correction? etc.). The second statement (which is missing some word, I believe) mentions background correction and comparison to the rest of the body (what is compared and how?).

Detailed description of the image analysis corresponding to the new Figure 4 is now provided in Materials and Methods section. The text corresponding to the former Figure 4 has also been amended with details as suggested by the referee, and has been moved to the Supplemental section along with former Figure 4.

Reviewers' Comments:

Reviewer #1 (Remarks to the Author):

The manuscript has been significantly improved, and the additional methods, experimental data and movies are informative and support their hypothesis. My queries on the initial manuscript have been addressed. Nevertheless there are errors in the preparation of the revised manuscript.

p. 9. 1. 1. What does “glassify” mean?

p. 9. 1. 17. Should be “Suppl. Fig. 5D”

p. 9. 1. 19. Should be “Suppl. Fig. 5C”

p. 16. 1. 30. “PopZ”

Movies 6 and 7 do not play unlike Moves 1 to 5 which play.

Several figures lack details in the legend.

Fig. 2.

Labels for “D”, “E” “F” and Z”G” are missing

Arrow for panel C, E, F line trace is not described in legend.

Panel D (Ci) DnaK+Ci data x 2 need 30oC and 37oC added.

Fig. 3. Resolution of panel B is low.

Fig. 6. What do “P”, “W” and “T” refer to ? Only “W” is in Figure 5.

Fig. 7. “...deviation is less than 10%”.

Suppl. Fig. 5. Panel C. “col E2”.

Suppl. Fig. 8. “W” is not shown in the figure?

Suppl. Movie 5 is not referenced in the appropriate section of the text on. p. 9 (Col. E2 expt). but on p. 10 with the ciprofloxacin expt.

Reviewer #2 (Remarks to the Author):

The authors have addressed my concerns from the initial manuscript. this version is significantly improved.

Reviewer #3 (Remarks to the Author):

The authors have addressed my concerns and I have no objection in accepting its publication.

Point by point response to reviewer comments, for clarity, reviewer comments are in italics:

Reviewer #1 (Remarks to the Author):

The manuscript has been significantly improved, and the additional methods, experimental data and movies are informative and support their hypothesis. My queries on the initial manuscript have been addressed. Nevertheless there are errors in the preparation of the revised manuscript.

p. 9. l. 1. What does “glassify” mean?

The term glassify was coined in the reference provided for this sentence (Parry et al. Cell 2014). We realise this may not have been clear, so we modified that section of the sentence to include a better description of glassification: « ... 2,4-dinitrophenol (DNP), an ATP uncoupler that glassifies the cytoplasm by reducing metabolic activity and increasing viscosity³³...»

p. 9. l. 17. Should be “Suppl. Fig. 5D”

p. 9. l. 19. Should be “Suppl. Fig. 5C”

p. 16. l. 30. “PopZ”

Thank you for highlighting these errors, we have corrected the text.

Movies 6 and 7 do not play unlike Moves 1 to 5 which play.

The movie files have been updated and re-checked prior to uploading on the Nature Communications server.

Several figures lack details in the legend.

More details, including the number of replicates and experimental repeats, have been added to the figure legends.

Fig. 2.

Labels for “D”, “E” “F” and Z”G” are missing

Arrow for panel C, E, F line trace is not described in legend.

Panel D (Ci) DnaK+Ci data x 2 need 30oC and 37oC added.

The figure and legend have been modified to address these omissions

Fig. 3. Resolution of panel B is low.

A higher resolution been uploaded for publication.

Fig. 6. What do “P”, “W” and “I” refer to ? Only “W” is in Figure 5.

The following sentence has been added to the figure legend: Representative fluorescence recovery kinetics of area corresponding to FRAP area (P), opposite pole (I) and whole bacterial body (W).

Fig. 7. "...deviation is less than 10%".

Thank you for pointing out the error, we have corrected the figure legend.

Suppl. Fig. 5. Panel C. "col E2".

Thank you for pointing out the omission of "E2", we have corrected the Figure.

Suppl. Fig. 8. "W" is not shown in the figure?

We have added the W to the figure.

Suppl. Movie 5 is not referenced in the appropriate section of the text on. p. 9 (Col. E2 expt). but on p. 10 with the ciprofloxacin expt.

Thank you for pointing out the error, we have corrected the text.

Reviewer #2 (Remarks to the Author):

The authors have addressed my concerns from the initial manuscript. this version is significantly improved.

Reviewer #3 (Remarks to the Author):

The authors have addressed my concerns and I have no objection in accepting its publication.